# Potential of Soil Stabilization Using Ground Granulated Blast Furnace Slag (GGBFS) and Fly Ash via Geopolymerization Method: A Review

**DOI:** 10.3390/ma15010375

**Published:** 2022-01-05

**Authors:** Syafiadi Rizki Abdila, Mohd Mustafa Al Bakri Abdullah, Romisuhani Ahmad, Dumitru Doru Burduhos Nergis, Shayfull Zamree Abd Rahim, Mohd Firdaus Omar, Andrei Victor Sandu, Petrica Vizureanu

**Affiliations:** 1Centre of Excellence Geopolymer and Green Technology (CEGeoGTech), Universiti Malaysia Perlis, Kangar 01000, Perlis, Malaysia; romisuhani@unimap.edu.my (R.A.); shayfull@unimap.edu.my (S.Z.A.R.); firdausomar@unimap.edu.my (M.F.O.); 2Faculty of Chemical Engineering Technology, Universiti Malaysia Perlis (UniMAP), Kangar 01000, Perlis, Malaysia; 3Faculty of Materials Science and Engineering, Gheorghe Asachi Technical University of Iaşi, Boulevard D. Mangeron, No. 51, 700050 Iasi, Romania; sav@tuiasi.ro (A.V.S.); peviz@tuiasi.ro (P.V.); 4Faculty of Mechanical Engineering Technology, Universiti Malaysia Perlis, Arau 02600, Perlis, Malaysia; 5Romanian Inventors Forum, Str. P. Movila 3, 700089 Iasi, Romania; 6Technical Sciences Academy of Romania, Dacia Blvd 26, 030167 Bucharest, Romania; 7Faculty of Civil Engineering, Mercu Buana Universiti, Jakarta 11650, Indonesia; syafwandi@mercubuana.ac.id

**Keywords:** geopolymer, fly ash, GGBFS, compressive strength, soil stabilization

## Abstract

Geopolymers, or also known as alkali-activated binders, have recently emerged as a viable alternative to conventional binders (cement) for soil stabilization. Geopolymers employ alkaline activation of industrial waste to create cementitious products inside treated soils, increasing the clayey soils’ mechanical and physical qualities. This paper aims to review the utilization of fly ash and ground granulated blast furnace slag (GGBFS)-based geopolymers for soil stabilization by enhancing strength. Previous research only used one type of precursor: fly ash or GGBFS, but the strength value obtained did not meet the ASTM D 4609 (<0.8 Mpa) standard required for soil-stabilizing criteria of road construction applications. This current research focused on the combination of two types of precursors, which are fly ash and GGBFS. The findings of an unconfined compressive strength (UCS) test on stabilized soil samples were discussed. Finally, the paper concludes that GGBFS and fly-ash-based geo-polymers for soil stabilization techniques can be successfully used as a binder for soil stabilization. However, additional research is required to meet the requirement of ASTM D 4609 standard in road construction applications, particularly in subgrade layers.

## 1. Introduction

Low-strength soil layers are frequently encountered in road construction, and they have a significant impact on various phases of construction design [1,2,3]. Another issue in civil engineering is clayey soil [4,5]. Clayey soil is a global issue, causing several difficulties for civil engineers, building enterprises, and property owners [6,7,8]. Clayey soils are seen as a potential natural hazard capable of wreaking havoc on engineering buildings [9,10]. Additionally, structures constructed on clayey soils have incurred significant damage as a result of the clayey soil’s undesirable and unpredictable characteristics [11,12,13].

According to ASTM D 2487 [14], the soil is commonly considered to have clay tendencies (the soil as to be classified as clay soil with high plasticity) when its liquid limit is greater than 50% and plasticity index is higher than 17%. Clays or soft soils primarily belong to the fine-grained group of soils and are classified as “clays” having the ability to change in volume when they come in contact with water [14]. Generally, the soil shrinks as the water content decreases and expands as the water content increases, due to the fact that the primary clay minerals have the potential to interact and absorb water [13,14,15]. This results in a high liquid limit and plasticity index [9,10,13,15]. Thus, clayey soil exhibits high swelling, shrinkage, and plasticity characteristics [14,15]. However, fine-grained soils were considered suitable applicants for stabilization [13,14,15].

In order to stabilize soil, the most common or usual approach was by removing the soft soil first [3,16,17,18]. More substantial materials, such as crushed rock, will replace soft soil [4,19]. Various researchers found another method to encounter this problem since the cost involved in replacing the materials was relatively high [18,20,21]. The mechanical and chemical stabilization alteration of one or more soil properties has been termed as soil stabilization [21,22]. Most researchers studied to improve the engineering properties by increasing the compressive strength of the soil [6,8,12,16,17,23] according to the ASTM D 4609 standard [24] (>0.8 Mpa). Soil stabilization can be described as a collective term for any physical, chemical, or combination of those approaches used to enhance certain features of natural soil in order for it to meet the engineering requirements [3,4,9,10,11,15,21,24].

Mechanical stabilization involves a physical process in altering the soil’s physical nature [25,26]. It entails compacting the soil in order to change its resistance, compressibility, permeability, and porosity [27,28]. Mechanical stabilization of a material is typically accomplished by adding another substance, which is fly ash or GGBFS, that enhances the grading or reduces the plasticity of the original material [28,29,30]. The original material’s physical qualities will be altered, but no chemical reaction will occur [26,31]. This method could increase the density of soil due to the elimination of maximum air. The material was not affected due to the particle size distribution [25,30,32]. However, the redistribution of the particles changes the structure [27,31]. Mechanical stabilization is frequently the most cost-effective method of enhancing the quality of low-grade materials (problematic soil) [26,27,29,33]. The stiffness and strength of the material will typically be less than those obtained through chemical stabilization and will frequently be unsuitable for heavy traffic pavements [33,34,35]. Additionally, a stabilizing agent may be required to improve the final qualities of the blended substance [30,34].

Other than mechanical soil stabilization, chemical soil stabilization is the most popular technique for remediation of poor ground conditions [19,36,37,38]. It is possible to alter the compressive strength, swelling potential, and volume change properties of soil through chemical stabilization processes such as mixing with ground granulated blast furnace slag (GGBFS) and fly ash by-products, as well as mixes of any of these materials [39,40,41,42,43]. The chemical substances work as compaction aids, binders, and water repellents and modify the soil behavior [44,45]. Stabilization of soils using chemical additives displays the usefulness of this technique, in engineering, such as road constructions and foundation substructure development [34,35,36,38,46].

The manufacturing of conventional soil stabilizers such as cement and lime results in considerable CO_2_ and energy emissions [47,48,49]. Therefore, civil engineering firms are continuously on the search for the new soil stabilizer, a low-carbon, sustainable substance, to be employed to replace cement as a soil stabilizer [47,50,51,52,53,54,55]. Geopolymerization is the process of polymerizing inorganic natural materials to form geopolymers [43,56,57]. Geopolymers have recently been shown to be a viable alternative to Portland cement for reinforcing degraded soils [52,53,54,55]. Geopolymers have outstanding engineering properties, such as increased strength and improved soil adhesion [36,40,56]. Materials containing high proportion of alumina (Al) and silica (Si) are required to manufacture this geopolymer material [47,48,57]. The Si and Al minerals found in industrial by-products such as fly ash from coal combustion and GGBFS from iron combustion are employed in geopolymer processes for soil stabilization [57,58,59,60]. In order to produce geopolymer, the sources of alumina and silica (fly ash and GGBFS) act as precursors that are easily dissolved in alkaline solutions resulted from alkaline activation, making geopolymerization possible [47,49,57,58,60,61,62,63,64,65,66,67,68].

In this paper, the usage of a GGBFS and fly ash Class-C-based geopolymer for soil stabilization by means of an unconfined compressive strength (UCS) test is reviewed and discussed.

## 2. Soil Stabilization by Conventional Method

### 2.1. Soil Stabilization Using Fly Ash

Fly ash is a material that is increasingly being used as a cement alternative in concrete mixtures and for soil stabilization [25,27,29,36,38,40,42,54]. The addition of fly ash is one of the methods to stabilize soil because fly ash is a pozzolan, in which it can bind to minerals in soil and make the soil stable, so as to reduce the swell shrinkage of the soil and improve soil strength [41,42,44,46,54,69,70,71,72,73]. The morphology of fly ash can be seen in Figure 1.

### 2.2. Soil Stabilization Using Ground Granulated Blast Furnace Slag (GGBFS)

Ground granulated blast furnace slag (GGBFS) is a by-product the manufacture of iron [27,35,36,40,41,42,43,47,54,64]. It is composed primarily of lime, alumina, and silicate [48,54,65,66,67]. GGBFS material is also used as a substitute for cement in concrete mixtures and as a material for soil stabilization [42,48,54,59,65,66]. Mixing GGBFS with soil improves compressive strength, permeability, and durability [39,54,65,66,67,68,69]. The morphology of GGBFS can be seen in Figure 2.

### 2.3. Strength of Soils after Stabilization with Fly Ash and Ground Granulated Blast Furnace Slag (GGBFS)

The strength of soil is a measure of its ability to absorb forces without collapsing [6,7]. A soil’s ability to bear normal and shear pressures can be used to determine its strength [65,71,72]. Apart from the primary reaction products generated, additional factors may impact the rate of rising in soil strength [51,73]. The presence of soil-borne accelerating or retarding chemical components may result in an increase in soil strength [65,66,68,74,75]. The summary of previous studies on soil stabilization application is presented in Table 1.

Research by Simatupang et al. [68] investigated the stability of soil with the addition of fly ash. Fly ash percentages range from 5–25% by dry weight of soil. By increasing the fly ash content in the samples and the curing time, the compressive strength value for fly ash increased. However, a long curing time is required to reach the optimum strength. To shorten the curing period, adding other materials such as GGBFS may be necessary.

Another study in 2016 by Dayalan et al. [65] investigated soil stabilization with ground granulated blast furnace slag (GGBFS). Different percentages of GGBFS 5–25% by dry weight of soil were used to stabilize the clayey soil. Based on the strength performance test, the optimum amount of GGBFS was determined to be 20%. Moreover, the result indicates that the inclusion of GGBFS increases the strength of clayey soils but the acquired properties do not meet the ASTM D 4609 soil-stabilizing criteria for road construction applications [76].

Another study in the same year by Mandal et al. [54] investigated soil stabilization using ground granulated blast furnace slag (GGBFS) and fly ash. Different samples were prepared with different proportions of soil, GGBFS, and fly ash. Based on the result, the best compressive strength values were obtained in a 10% GGBFS and 10% fly ash mixture. This provides proof that the addition of GGBFS and fly ash can improve the clayey soil’s mechanical properties. However, the soil strength value still does not comply the ASTM D 4609 standard [76], which requires a value greater than 0.8 MPa. To comply with the requirement, increasing the percentage of fly ash and GGBFS mixture proportions may be required.

Research performed in 2019 by Neeladharan et al. [39] investigated the possibility of stabilizing expansive soils through the use of a binder comprising of fly ash and ground granulated blast furnace slag (GGBFS).The clayey soil was mixed with different percentages fly ash of 5–25% and GGBFS of 2.5–10% by dry weight of soil. According to the results of the unconfined compressive strength test, a binder percentage of 20% is recommended as the optimal. However, the unconfined compressive strength value did not fulfill the ASTM D 4609 standard [76], which must be greater than 0.8 MPa. To fulfill the standard, increasing the percentage of fly ash mixture proportions and adding other ingredients such as GGBFS may be required.

Another study in 2014 by Oormila et al. [66] investigated the potential of using GGBFS as a stabilizer for the clay/soft soil. The soft soil was mixed with GGBFS at different percentages (15–25% by dry weight of soil) with curing times of 7, 14, and 21 days. The result indicates that the use of GGBFS increased the strength characteristics of the soil. Based on compressive strength, the optimum amount of GGBFS was 20%, as it increased the strength by about 73.79% of clayey soil. This provides proof that the GGBFS can improve the strength of the clayey soil. However, a long curing time is required to reach optimum strength. To reduce the curing time and increase strength, it may be essential to combine two types of precursors (fly ash and GGBFS) and increase the percentage of fly ash mixture proportions.

Research performed by Sharma et al. [67] investigated the possibility of utilizing a binder composed of fly ash and powdered granulated blast furnace slag to stabilize expansive soils (GGBFS). The expansive soil was mixed at different percentages of fly ash 70% and GGBFS 30% with curing times of 7, 14, and 28 days. Based on the strength result, the strongest soil was achieved after 28 days of curing time, with a compressive strength value of 0.45 MPa. However, the unconfined compressive strength value does not fulfill the ASTM D 4609 standard [76], which must be more than 0.8 MPa. Furthermore, a long curing time is required to reach optimum strength. To shorten the curing period and increase compressive strength, increasing the percentage of fly ash and GGBFS mixture proportions may be required.

Another study by Tyagi et al. [69] investigated soil stabilization with GGBFS and fly ash. Fly ash and GGBFS were utilized in amounts of 0%, 5%, 10%, 15%, 20%, 25%, and 30%, respectively, by weight of the soil sample. The strength value increases as the amounts of fly ash and GGBFS reach their maximum values of 18% and 30%, respectively. However, the unconfined compressive strength value does not fulfill the ASTM D 4609 standard [76], which must be greater than 0.8 MPa. To reduce the curing time and increase strength, it may be essential to combine two types of precursors (fly ash and GGBFS) and increase the percentage of fly ash mixture proportions.

Research performed by Mujtajab et al. [70] investigated the enhancement of expansive soils’ engineering qualities with the addition of GGBFS. The impact of GGBFS in stabilizing these expansive soils was examined by applying various amounts of GGBFS between 0% and 55% to these soil samples. The strength of a remolded sample after 28 days was increased by approximately 35% with the addition of 30% GGBFS. Although, the compressive strength of the soil fulfils the ASTM D 4609 [76] standard, it requires a long curing time to get the optimum strength value. To reduce the curing time and increase strength, adding other materials such as fly ash may be necessary.

Therefore, additional research is necessary to determine the possibility of employing GGBFS and fly ash as soil stabilizers in increasing soil compression strength and shortening the curing time in order to maximize soil power. Geopolymers have recently been shown to be an effective replacement to Portland cement for reinforcing degraded soils; hence, geopolymers can be pushed for their suitability for use as a concrete substitute [52,53,54,55].

## 3. Geopolymer

Geopolymers can usually be synthesized from many materials with high concentration of aluminosilicates. Geopolymer precursors that are high in silica (Si) and alumina (Al) minerals, such as fly ash and GGBFS, are highly suggested for geopolymerization in soil stabilization applications [48,51,54,66]. All the aluminosilicate materials must be activated by a second raw material known as alkali activator solution [56].

Geopolymerization is an integrated process for synthesizing geopolymers, which involves leaching, diffusion, reorientation, polymerization, and condensation [56]. Three stages of polymerization occur: (1) dissolution of oxide minerals from source materials (typically silica and alumina) under extreme alkaline conditions; (2) orientation of dissolved oxide minerals followed by gelation; and (3) polycondensation to form a three-dimensional network of silico-aluminates structures [56]. Duxson et al. [56], proposed a polymerization that process involves several steps proposed in the conceptual model. A general mechanism of geopolymerization is shown in Figure 3.

This model assumes that geopolymerization starts with the dissolution of the source materials by the alkali solution, which causes the breaking of the aluminosilicate bond and releases silica and alumina, mainly in the source materials [78,79]. The aluminosilicate chain’s negative charge is balanced by alkali cations such as potassium, sodium, or calcium. Thus, the silica and alumina content in the source material has a significant effect that governs geopolymer performance [73].

Furthermore, the dissolution rate increases as the solution alkalinity increases. This rate controls the time required to reach saturation, after which a supersaturated aluminosilicate solution is reached. Then, the main condensation process begins, and the aluminosilicate gel in the form of oligomers precipitates to produce larger and more stable networks [56].

The first polymer (Gel 1) is formed when the solution contains a higher Si and Al concentration. As the reaction continues, more Si enters the solution, resulting in gels containing higher amounts of Si (Gel 2). The initial setting starts when the condensation rate of the aluminosilicate species exceeds the dissolution rate. Finally, polycondensation and rearrangement processes continue to produce more connected 3D networks, forming the final geopolymer matrix [56].

### 3.1. Source of Raw Materials

The most common source material to produce geopolymers is fly ash [51,72,80,81,82,83,84,85]. High-strength geopolymers generally could be obtained from Class F fly ash (low calcium mineral) [48,54,66,86,87]. However, Class C fly ash (high calcium) has shown that it can also be used to produce geopolymer [11,50,87,88,89,90]. The fly ash from different sources could affect the final properties since they have different levels of reactivity under specific geopolymer synthesis conditions [48,51,76]. The chemical content, quantity of fly ash, and activator solution will influence fly-ash-based geopolymer properties in the fresh and hardened state [51,72,73].

In 1957, the first method to utilize a slag-based geopolymer as a binder in building was created [91]. Ground granulated blast furnace slag (GGBFS) is a granular glass composed primarily of calcium oxide (CaO), silicon dioxide (SiO_2_), aluminium oxide (Al_2_O_3_), and magnesium oxide (MgO) [76,77,78,79]. It is an amorphous by-product of the production of pig iron from iron ore, coke combustion residue, and fluxes such as limestone [48,54,75]. The existence of calcium (CaO) content in GGBFS contributed to the shortened setting time and development of the compressive strength of the clay soil [48,54,66,73,82,83,85,88]. The reaction between GGBFS and alkali activator solution forms a calcium–aluminate–silicate–hydrate (C–A–S–H) gel-forming within the geopolymer matrix. These hydration products, along with aluminosilicate structure in the GGBFS samples, contributed to significantly gaining high strength [78].

### 3.2. Alkali Activator

The most common alkali solution used in geopolymerization is a mixture of sodium silicate (Na_2_SiO_3_), and sodium hydroxide (NaOH) [48,51,72,82]. The sodium hydroxide (NaOH) solution concentration has a significant effect on the physical and mechanical properties of soil-stabilization-based geopolymer [48,51,72,75,82]. This solution contains hydroxide ions (OH-) and sodium ions (Na+), which initiate the reaction between the internal silicate (Si) and aluminate (Al) components, initiating the dissolving process [48,51,75,82]. Using a mixture of both sodium silicate (Na_2_SiO_3_) and sodium hydroxide (NaOH) as an alkali activator will give better strength than using a sodium hydroxide (NaOH) solution only [51,74,80,83]. The reaction of sodium silicate (Na_2_SiO_3_) to process polymerization is crucial in dissolving Si, and the mixing ratio of sodium hydroxide (NaOH) concentration is also crucial in producing good strength of the product from geopolymer [48,51,75,82].

Generally, higher strength could be obtained by using higher contents of NaOH and sodium silicate [48,75]. However, there will be an adverse effect on strength if there was too much alkali in the composition [51,82,83]. In order to control the compressive strength, various alkali activator compositions were usually used [48,51,72,82]. The use of a higher molar concentration of alkali ions could accelerate the reactants in the chain reaction [48,51,72,75,82]. Nevertheless, it might lead to the rapid loss inconsistency during the mixing process due to the faster reaction of the polymer [51,81,82]. Then, to provide the maximum mechanical properties, there should be an optimum alkali activator content [51,81,84].

### 3.3. Strength of Soil after Stabilization with Fly Ash and Ground Granulated Blast Furnace Slag (GGBFS) Geopolymer

Numerous investigations on geopolymers have been undertaken, which are used to make ceramics, earth bricks, mortar, and concrete. Stabilizing soil with geopolymer binders is a relatively recent concept. The use of ground granulated blast furnace slag and fly-ash-based geopolymers to stabilise clayey soil showed promising results. The summary of previous studies on soil stabilization with fly ash and GGBFS-based geopolymer is presented in Table 2.

Research by Anne et al. [72] investigated the use of fly ash in the synthesis of geopolymer for soil stabilization. Fly ash was used in proportions of 15% and 25% and ratio of alkali activator Sodium Silicate: Sodium Hydroxide:Sodium Aluminate of 50:50:0, 33:33:33, 50:20:30. According to the findings, strength increased as the amount of fly ash increased. However, the strength value still does not comply with the ASTM D 4609 standard [76], which requires a value greater than 0.8 MPa. Furthermore, a longer curing time is required to reach the optimum strength. To shorten the curing period and increase compressive strength, adding other materials such as GGBFS may be necessary.

In 2017, Abdullahet al. [51] investigated the effect of the alkaline activator/fly ash ratio on the stability of geopolymer-stabilized soil. A variety of mix designs were made and cured for 7 and 28 days at varied fly ash/alkaline activator ratios and Na_2_SiO_3_/NaOH ratios. The molecular weight of the geopolymer and the proportion of geopolymer in the soil were set to 10 molar and 8%, respectively. The maximum strength was attained with a fly ash/alkaline activator ratio of 2.5 and a Na_2_SiO_3_/NaOH ratio of 2.0 after 28 days of curing period. Additionally, the mixture of fly ash geopolymers contributed to filling the large surface area of the voids between the clay particles and controlling the moisture content of the clay. This can cause the clay to become stable and compact and increase the compressive strength of the clay. However, the results are not in accordance with the ASTM D 4609 [76] soil stabilization criteria for road construction applications. The strength of the soil should be greater than 0.8 MPa [76,91]. To comply with the standard, additional chemicals such as GGBFS may be required, as well as an increase in the percentage of fly ash mixture proportions.

In 2018, Parhi et al. [75] investigated the stabilization of soil with the use of an alkali-activated fly-ash-based geopolymer. The fly ash is activated using concentrations of sodium hydroxide of 10, 12.5, and 15 molars. The various percentages of fly ash (20–40%) relative to the expanding soil’s total solids are employed. The ratios of activator to ash (liquid to solid mass ratio) were maintained between 1 and 2.5. The 10 molal samples have a greater three- and seven-day strength than the 12.5 and 15 molar samples, which make them more cost effective than the 12.5 and 15 molal samples. However, the strength results do not adhere to the ASTM D 4609 [76] standard soil stabilization criteria for road construction applications. To comply with the requirement, increasing the percentage of fly ash mixture proportions and adding other ingredients such as GGBFS may be required.

In the same year, Leong et al. [81] investigated the strength improvement of soil stabilization with fly-ash-based geopolymer: an appraisal of soil, fly ash, alkali activators, and water. Molarity was fixed at 8 molars and the ratios of fly ash/soil were 0.3, 0.6, 0.8, and 0.9. According to the results, strength increases as the fly ash/soil ratio increases. Although the compressive strength of the soil fulfils the ASTM D 4609 [76] standard, it required a long curing time to obtain the optimum strength value. To shorten the curing period, increasing the percentage of fly ash mixture proportions and molarity and adding other materials such as GGBFS may be necessary.

In another study, Shihab et al. [80] investigated the influence of NaOH molar concentration on the mechanical strength of a soft clayey soil stabilized with a fly-ash-based geopolymer after initial heating. The fly ash was activated with 8, 10, 12 and 14 molars. Dosages of fly ash were selected as 8%, 10%, 12%, and 14% of dry weight of soil. Based on the result, the optimum molar concentration is 12 M. Although the compressive strength of the soil fulfils the ASTM D 4609 [76] standard, it requires a high molarity to obtain the optimum compressive strength value.

Another study conducted by Thomas et al. [73] investigated the stabilization of soils through the use of alkali-activated GGBFS. The GGBFS doses of 6%, 9%, 12%, 15%, and 20% of dry weight of soil were chosen. The optimal dosage for GGBFS can be set as 20% based on the strength. However, the strength results do not fulfill the ASTM D 4609 [76] standard soil stabilization criteria for road construction applications. To comply with the requirement, increasing the molarity and adding other ingredients such as fly ash may be required.

Another research performed by Phummiphan et al. [88] examined the use of geopolymer stabilized soil, fly ash, and GGBFS blends as a pavement base material. Molarity was fixed at 5 molars. Dosages of fly ash were selected as 30% and GGBFS of 10%, 20%, and 30%. The soil sample was prepared by mixing it with fly ash and GGBFS and curing it for 7, 28, and 60 days. Based on unconfined compressive strength (UCS) result, the optimum dosage can be selected as 20% for GGBFS and 30% for fly ash. However, the compressive strength value still did not comply with the ASTM D 4609 standard [76], which requires a value greater than 0.8 MPa. Furthermore, a long curing time is required to reach optimum strength. To shorten the curing period and increase compressive strength, increasing the percentage of fly ash mixture proportions and molarity may be necessary. Additionally, the existence of calcium (CaO) content in GGBFS contributed to the shortened setting time and development of the compressive strength of the clay soil. This finding was supported by a previous study by Aziz et al. [78] where the reaction between GGBFS and alkali activator solution formed a calcium–aluminate–silicate–hydrate (C–A–S–H) and calcite (CaCO_3_) within the geopolymer matrix. These hydration products, along with aluminosilicate structure in the GGBFS samples, contributed significantly to high strength gain.

In addition, a few crucial factors influencing the properties of geopolymer so as to contribute to achieving a good mix of design and formulation of geopolymer include solid-to-liquid (S/L) ratio, sodium hydroxide molarity, and sodium silicate (Na_2_SiO_3_)/sodium hydroxide (NaOH) ratio, which will be explained in the next section.

## 4. Factors Affecting the Geopolymer Properties

Commonly, in soil stabilization-based geopolymer, fly ash Class C and GGBFS are used as the primary source material containing mostly high content in silica (Si) and alumina (Al) that dissolve in alkali solution for geopolymerization in soil stabilization application [51,65,72,73,74]. A few crucial factors influencing the properties of geopolymer so as to contribute to achieving a good mix of design and formulation of geopolymer include solid-to-liquid (S/L) ratio, sodium hydroxide molarity, and sodium silicate (Na_2_SiO_3_)/sodium hydroxide (NaOH) ratio [51,74,92,93,94].

### 4.1. Effect of Solid to Liquid Ratio on Geopolymer

The solid-to-liquid (S/L) ratio corresponds to the aluminosilicate source to activator solution ratio [72,73,88,92,95,96]. The aluminosilicate source’s alkali activator is a blend of solid and liquid. The liquid is extremely alkaline, and the solid contains an appropriate amount of highly reactive silicate aluminate [40,48,50,51,57,78]. Based on a previous study by Alonso et al. [97], it is said that the initial solid content highly influences the rate of geopolymer formation; it was evident that a large number of precipitates was observed with an increasing solid-to-liquid ratio. This is said to be due to high dissolved reactant species.

As the solid-to-liquid ratio rose, the geopolymers sample became less homogeneous due to the limited amount of alkali activator [48,51,72,74,78,93,94,98]. At this solid-to-liquid (S/L) ratio, the liquid content was negligible in comparison to the solid content [51,74]. Thus, it resulted in a soil-based geopolymer with low compressive strength. Abdullah et al. [51] also analyzed the influence of solid-to-liquid (2.0, 2.5, and 3.0) ratio to soil stabilization using fly-ash-based geopolymer, observing a lower extent of binder formation, which resulted in a soil and fly-ash-based geopolymer sample with a solid-to-liquid ratio higher than 2.5. This is due to the high degree of supersaturation of the aluminosilicate phase, so as to produce a less connected geopolymer structure [51,74].

### 4.2. Effect of Sodium Hydroxide Molarity on Geopolymer

The concentration of sodium hydroxide (NaOH) solution is a critical parameter that influences the geopolymer’s properties. This solution is one of the alkali activators used in the production of geopolymer [9,40,48,50,51,57,74,78,99]. It contains and provides hydroxide ion (OH-) and sodium ion (Na+), mainly responsible for the dissolution of aluminosilicate source materials and the polymerization process of geopolymer [74,78,82,83,99,100,101]. The molarity of NaOH solution is reported to significantly affect the workability, geopolymerization reaction, and strength development of the final product [74,87,102,103]. The obstacle of using NaOH solution in the geopolymer synthesis is the low strength development if the molarity used is too high or too low [71,72,74,80,92,94].

There were a few studies on the effect of NaOH molarity on the geopolymer [72,74,75,80,104]. According to Malkawi et al. [105], a strong alkali is needed due to the alkali activator’s function, which is required to partially or entirely dissolve the silica and alumina available in the source materials as explained in details in the polymerization mechanism [74,78,101,106]. The researcher found that 10 M NaOH solution is suitable for Class F and Class C fly-ash-based geopolymer, respectively [51,74,101,107,108].

### 4.3. Effect of Sodium Silicate to Sodium Hydroxide Ratio on Geopolymer

Previous research has employed the merger of sodium hydroxide and sodium silicate as an alkali activator [48,51,74,83,88,106,107,108,109,110]. As reported by Tempest et al. [111], the degree of reactivity of raw geopolymer material decreases when the sodium silicate (Na_2_SiO_3_)/sodium hydroxide (NaOH) ratio is too high. This is because sodium hydroxide solution plays an essential role in dissolving the raw materials [111]. However, higher sodium silicate (Na_2_SiO3)/sodium hydroxide (NaOH) ratios are encouraged to be studied for geopolymer systems due to commercial and marketing interest, lowering the workability and strength of the mixture [51,74,78,102,103,109,110]. This is supported by multiple another study, which also found that sodium silicate (Na_2_SiO_3_)/sodium hydroxide (NaOH) ratio at 2.0 in soil stabilization using fly-ash-based geopolymer process shows the best performance [51,74].

Much attention has been given to the utilization of stabilization materials for soil stabilization, but only a few scholars investigated the use of geopolymer for soil stabilization application. Instead of focusing on stabilizing soil with fly ash and GGBFS-based geopolymer, this paper investigated the soil stabilization using fly ash and GGBFS via the geopolymerization process, where the soil stabilization was performed by mixing the soil, fly ash, and GGBFS directly with alkali solutions. In other words, the soil, fly ash, and GGBFS act as the source materials for polymerization, producing soil-based geopolymers.

## 5. Summary and Future Works

This paper presents the clayey soil or problematic soil that can be stabilized and the strength improved by using a fly ash Class C and GGBFS-based geopolymerization process. The experimental results indicate that stabilization using stabilizers chemically and mechanically alters the majority of clayey soils, resulting in a significant increase (>0.8 Mpa) in unconfined compressive strength (UCS). Geopolymerization process is a relatively new approach for soil stabilization that has the potential to outperform previous treatments (conventional method).

Additionally, this paper discusses the factor that determines the geopolymer’s properties. Two factors have been shown to have a major effect on the geopolymer’s properties: the S/L ratio and the Ca concentration of the geopolymer. The S/L ratios vary according to the materials utilized. The majority of assessments of fly ash Class C and GGBFS utilized an S/L ratio between 1 and 3. Thus, analyzing prior research will aid in determining the optimum S/L ratio to be applied in future studies. On the other hand, the calcium concentration of a geopolymer can be connected to the materials employed. For instance, increasing the amount of fly ash Class C and GGBFS used will improve the calcium content.

While a wealth of literature is available on using fly ash Class C and GGBFS-based geopolymers, they nearly always refer to their use in building materials. Several researchers reported on the use of geopolymers for soil stabilization but did not discuss their applicability in road construction, particularly in the subgrade layer. Additionally, if the soil stabilization only uses geopolymers, it does not produce CO_2_. Hence, it is impossible to evaluate and discuss the CO_2_ emission reduction ratio of soil-based geopolymers. However, if soil stabilization involves the use of materials such as cement, it is possible to evaluate and discuss the CO_2_ emission because one ton of cement is manufactured, accounting for approximately one ton of greenhouse gas CO_2_ released into the atmosphere as a result of lime decarbonization in the kiln during cement manufacture. Moreover, the swell behavior, flexural strength, or abrasion resistance of soil stabilization-based geopolymer using GGBFS and fly ash Class C in road construction, particularly in the subgrade layer of the soil, have not been investigated. Thus, this article only discusses the performance of using geopolymer technology in soil stabilization based on compressive strength in road construction applications, especially in subgrade layers. In addition, the potential applications of GGBFS-FA geo-polymer are seen in soil stabilization operation with deep injection techniques, slope stabilization techniques, and in situ stabilization techniques. Based on the identified gaps, several future works are proposed in this study as listed below:i.Previous research has demonstrated that soil stabilization based on geopolymers using fly ash Class C and GGBFS as raw materials can increase the compressive strength of clayey soil. Thus, it is advised that future works concentrate on using fly ash Class C and GGBFS geopolymers as soil-stabilizing materials.ii.The mix design of soil stabilization-based geopolymer is critical in defining the mechanical and physical properties of soil stabilization-based geopolymer. Thus, the optimal solid-to-liquid ratio, sodium hydroxide to sodium silicate ratio, and sodium hydroxide molarity must be further researched in relation to soil-stabilizing requirements such as unconfined compressive strength and Atterberg limits test.iii.The effects of various curing temperature on soil-based geopolymer need to be further investigated.

## Figures and Tables

**Figure 1 materials-15-00375-f001:**
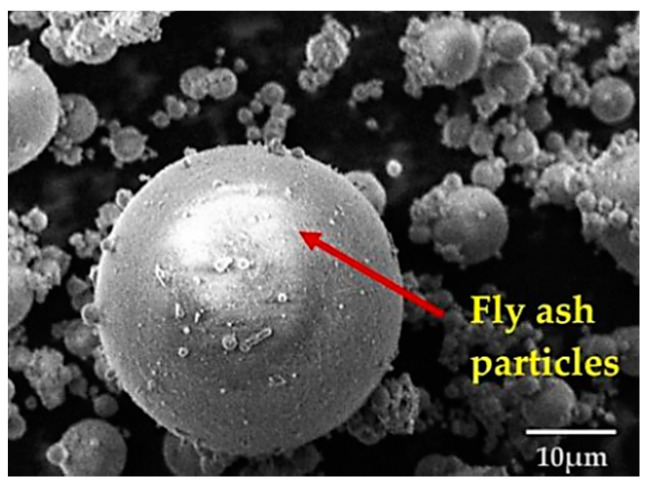
The microstructure image at 2000× magnification of fly ash [74].

**Figure 2 materials-15-00375-f002:**
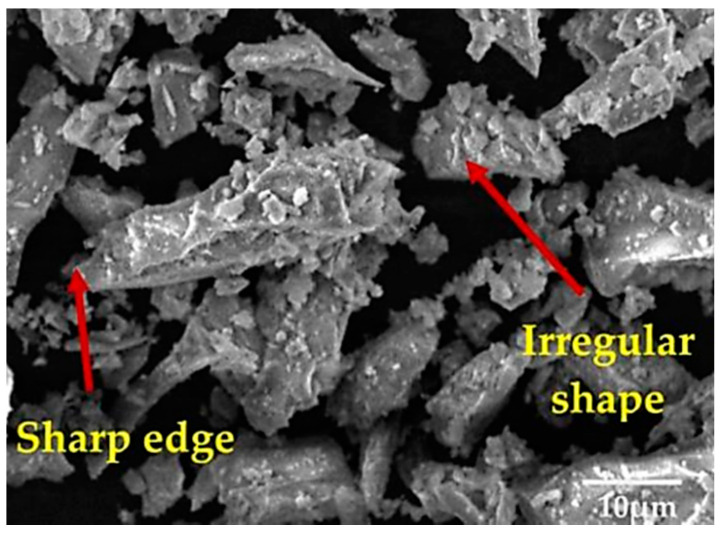
The microstructure image at 2000× magnification of GGBFS [74].

**Figure 3 materials-15-00375-f003:**
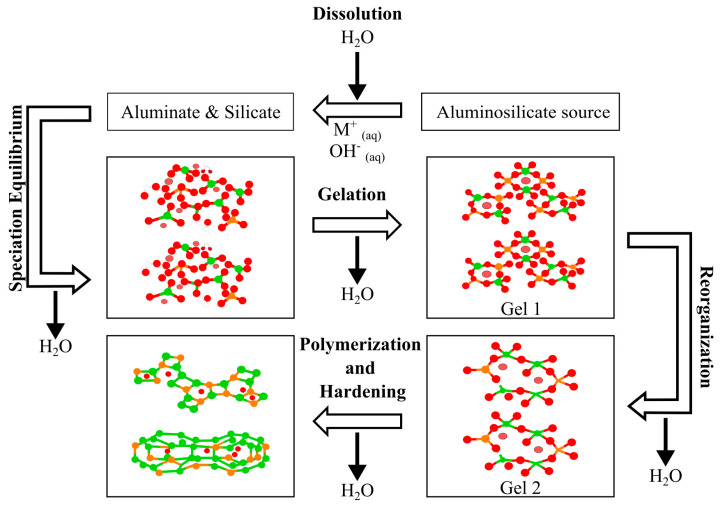
Geopolymerization mechanism.

**Table 1 materials-15-00375-t001:** The summary of previous studies on soil stabilization application.

No	Author	Testing	Raw Materials	Percentage of Blended Mix Proportion (%)	Curing Condition	Finding
1.	Simatupang et al. [68]	Unconfined compressive strength (ASTM D 2166) [77]	Fly ash	Fly ash: 5%, 10%, 15%, 20%, and 25%	7, 14, 28, and 56 days curing at room temperature	By increasing the fly ash quantity in the specimen and curing period, the strength of fly ash increased.
2.	Dayalan J et al. [65]	Unconfined compressive strength (ASTM D 2166) [77]	GGBFS Fly ash	Fly ash: 5%, 10%, 15%, 20%, and 25% GGBFS: 5%, 10%, 15%, 20%, and 25%	1 day curing	The optimum compressive strength value for fly ash is 15% and GGBFS is 20%, respectively.
3.	Neeladharan et al. [39]	Unconfined compressive strength (ASTM D 2166) [77]	GGBFS Fly ash	Fly ash: 5%, 10%, 15%, and 20%, GGBFS: 2.5%, 5%, 7.5%, and 10%,	1 day curing	The strength value increases with an increase in amount of fly ash and GGBFS, which attained maximum value at 15% and 10%, respectively.
4.	Oormila et al. [66]	Unconfined compressive strength (ASTM D 2166) [77]	GGBFS Fly ash	Fly ash: 5%, 10%, 15%, and 20%, GGBFS: 15%,20%, and 25%,	7, 14, and 21 days curing at room temperature	The optimum compressive value for fly ash is 10% and GGBFS is 20%, respectively.
5.	Sharma et al. [67]	Unconfined compressive strength (ASTM D 2166) [77]	Fly ash GGBFS	Fly ash: 70% GGBFS: 30%	7, 14, and 28 days curing at room temperature	The optimum soil strength value is 0.45 MPa.
6.	Mandal et al. [54]	Unconfined compressive strength (ASTM D 2166) [77]	GGBFS Fly ash	Fly ash: 5%, 10%, 15%, 20%, and 25% GGBFS: 10%	1 day curing	The maximum value was found at 10% GGBFS and 10% fly ash, which is 4.51 kg/cm^2^
7.	Tyagi et al. [69]	Unconfined compressive strength (ASTM D 2166) [77]	GGBFS Fly ash	Fly ash: 0%, 3%, 6%, 9%, 12%, 15%, and 18%, GGBFS: 0%, 5%, 10%, 15%,20%, 25%, and 30%	7 and 14 day curing	The strength value increases with increases in amount of fly ash and GGBFS, which attained maximum value at 18% and 30%, respectively.
8.	Mujtaba et al. [70]	Unconfined compressive strength (ASTM D 2166) [77]	GGBFS	GGBFS: 5%, 10%, 15%, 20%, 30%, 40%, 50%, and 55%	0, 3, 7, 14, and 28 days curing at room temperature	The optimum compressive strength value for GGBFS is 30%.

**Table 2 materials-15-00375-t002:** The summary of previous studies on soil stabilization with fly ash andground granulated blast furnace slag (GGBFS)-based geopolymer.

No	Author	Testing	Raw Materials	Activator Chemical	Molarity NaOH (M)	Percentage of Blended Mix Proportion (%)	Curing Condition	Finding
1.	Anne et al. [72]	Unconfined compressive strength (ASTM D 2166) [77]	Fly ash	Sodium silicate (Na_2_SiO_3_) Sodium hydroxide (NaOH)	Na/Al: 2.05 Si/Al: 2.64 Na/Si: 0.78	Fly ash: 15% and 25%	7, 14, and 28 days of curing at room temperature	Adding more fly ash increased compressive strength.
2.	Thomas et al. et al. [73]	Unconfined compressive strength (ASTM D 2166) [77]	GGBFS	Sodium silicate (Na_2_SiO_3_) Sodium hydroxide (NaOH)	1 M	GGBFS: 6%, 9%, 12%, 15%, 20%, and 30%	7 and 28 days of curing at room temperature	The optimal dose for GGBFS is 20%.
3.	Abdullah et al. [51]	Unconfined compressive strength (ASTM D 2166) [77]	Fly ash	Sodium silicate (Na_2_SiO_3_) Sodium hydroxide (NaOH)	10 M	Fly ash: 8%	7 and 28 days of curing at room temperature	The optimum strength obtained at the fly ash/alkaline activator ratio 2.5 and Na_2_SiO_3_/NaOH ratio 2.0 at 28 days of the curing period.
4.	Parhi et al. [75]	Unconfined compressive strength (ASTM D 2166) [77]	Fly ash	Sodium silicate (Na_2_SiO_3_) Sodium hydroxide (NaOH)	10 M, 12.5 M and 15 M	Fly ash: 20%, 30%, and 40%	3 and 7 days of curing at room temperature	10 molal samples provide greater 3 and 7 strength than 12.5 and 15 molal samples.
5.	Phummiphan et al. [88]	Unconfined compressive strength (ASTM D 2166) [77]	Fly ash GGBFS	Sodium silicate (Na_2_SiO_3_) Sodium hydroxide (NaOH)	5 M	Fly ash: 30% GGBFS: 10%, 20%, and 30%,	7, 28, and 60 days of curing	The optimal dosage is 20% for GGBFS and 30% for fly ash.
6.	Leong et al. [81]	Unconfined compressive strength (ASTM D 2166) [77]	Fly ash	Sodium silicate (Na_2_SiO_3_) Sodium hydroxide (NaOH)	8 M	Ratio Fly ash/Soil: 0.3, 0.6, 0.8, and 0.9	1 day of curing at 100 °C temperature	The compressive strength improves as fly ash/soil ratio increases.
7.	Shihab et al. [80]	Unconfined compressive strength (ASTM D 2166) [77]	Fly ash	Sodium silicate (Na_2_SiO_3_) Sodium hydroxide (NaOH)	10 M, 12M, and 14M	Fly ash: 8%, 10%, 12%, and 14%	1 day of curing at 70 °C temperature	The optimum molar concentration is 12 M.

## Data Availability

Not applicable.

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
