# Peer review of "Potential of Soil Stabilization Using Ground Granulated Blast Furnace Slag (GGBFS) and Fly Ash via Geopolymerization Method: A Review"

_materials, 2022, doi:10.3390/ma15010375_

Round 1
Reviewer 1 Report
Journal: Materials MDPI
Manuscript ID 1518891
Manuscript Type: Review Paper
Title: Potential of Soil Stabilization using Ground Granulated Blast Furnace Slag (GGBFS) and Fly Ash via Geopolymerization Method: A Review
Recommendation: revisions
GENERAL COMMENT. The authors presented an interesting review article on the use of ground granulated blast furnace slag and fly-ash geopolymerization. The motivation/justification given by the authors to publish this review article does not appear well defined in the abstract. In general, the article is well structured and written, but doesn’t introduce a novelty about the models to predicting GGBFS-Fly ash geopolymerization strength. The reviewer recommends requesting revisions subjects to the authors reflecting on the specific comments detailed below.
SPECIFIC COMMENTS.
-Abstract. The authors already begin by highlighting the objective of the article. The authors should reflect on the motivation and novelty in the abstract. In addition, some numerical results should also be highlighted.
- How could the suction have influenced the strength of GGBFS-Fly ash geopolymers? were there variations in the moisture of the specimens from molding to testing day?
-Relate some important properties of GGBFS and fly ash. Density? granulometry? chemical composition? morphology? How could these properties affect resistance and geopolymerization? Please, add some photos of raw materials and compacted materials. Add a flowchart of alkaline activator of GGBFS-FA
- various models to predict the strength of GGBFS-fly ash were introduced in the literature. Please, consider some papers to describe this model inside the introduction and discussions
-Based on the results, what is the best binder (or alkaline activator) for stabilizing GGBFS-Fly ash? (Considering the strength, compaction effort, and curing time??)
-Please relate some applications of GGBFS-FA geopolymers (geomaterials, pavement, rammed earth, etc??)
-Conclusions. Use more numerical results for concluding.
Author Response
Dear Reviewer,
Thanks for your comments and suggestions.
|
No. |
Comments |
Response |
|
1 |
English language and style (x) English language and style are fine/minor spell check required |
Has been sent for proof read again. |
|
2 |
Does the introduction provide sufficient background and include all relevant references? Comments: Yes |
Thanks for your comments. |
|
3 |
Is the research design appropriate? Comments: Yes |
Thanks for your comments. |
|
4 |
Are the methods adequately described? Comments: Yes |
Thanks for your comments. |
|
5 |
Are the results clearly presented? Comments: Can be improved |
Has been improved. Please refer to the Sections 3 and 4, Lines 220-441. All changes are highlighted with the blue colour fonts. |
|
6 |
Are the conclusions supported by the results? Comments: Can be improved |
Has been improved. Please refer to the Pages 14-15, Lines 443-488. |
|
8 |
Abstract. The authors already begin by highlighting the objective of the article. The authors should reflect on the motivation and novelty in the abstract. In addition, some numerical results should also be highlighted. |
Thanks for your comments. The abstract has been revised and improved accordingly. Please refer to the abstract section, page 1, lines 29-36. |
|
9 |
How could the admixture have influenced the strength of GGBFS-Fly ash geopolymers? were there variations in the moisture of the specimens from moulding to testing day? |
Thanks for your comments. From a mixture of fly ash- GGBFS geopolymers contribute to filling the large surface area of the voids between the clay particles and controlling the moisture content of the clay. This can cause the clay to become stable, compact and increase the compressive strength of the clay. The optimum strength is expected to be obtained on the 7- 28 test day. Please refer to the section 3.3, page 12, Lines 315-328. |
|
10 |
Please, add some photos microstructure of raw material. Add a flowchart of alkaline activator of GGBFS-FA Geopolymer. |
Thanks for your comments and suggestions The microstructure of raw materials and flowchart of alkaline activator of GGBFS-FA Geopolymer has been added accordingly. Please refer to the section 2, page 3, Lines 116-126. Please refer to the section 3, page 8, Line 235. Please refer to the section 3.2, page 9, Lines 268-273. |
|
11 |
Various models to predict the strength of GGBFS-fly ash were introduced in the literature. Please, consider some papers to describe this model inside the introduction and discussions. |
Thanks for your comments and suggestions. To predict the strength of fly ash-GGBFS, the unconfined compressive strength (UCS) test model is used as a practical indicator. Please refer to the section 3.3, pages 12-13, Lines 307-377. |
|
12 |
Based on the results, what is the best binder (or alkaline activator) for stabilizing GGBFS-Fly ash? |
Thanks for your comments. The best binder for stabilizing GGBFS-Fly ash is a mixture of sodium silicate (Na2SiO3), and sodium hydroxide (NaOH). The reaction of sodium silicate (Na2SiO3) to process polymerization is crucial in dissolving and reformulating more (Si), and the mixing ratio of sodium hydroxide (NaOH) concentration is also crucial in producing good strength of the product from geopolymer. This explanation has been added in the new manuscript. Please refer to the Page 9, Lines 275-294. |
|
13 |
Please relate some applications of GGBFS-FA geopolymers. |
Thanks for your comments. The applications of GGBFS-FA geopolymers have been added in this new manuscript. Please refer to the Summary and future works section, Page 15, Lines 473-475. |
|
14 |
Conclusions. Use more numerical results for concluding. |
Thanks for your comments and suggestions. The Summary and Future Works Section has been revised and improved accordingly. Please refer to the Summary and future works section, pages 14-15, lines 443-475. |
Reviewer 2 Report
The present study attempted to summarize the published studies on the usage of fly ash and ground granulated blast furnace slag as the geopolymerization precursors. The idea is satisfying, but the presentation of the conclusions is at a low level. The paper must be rephrased, and also many figures can be included to clearly show the differences in published works. One very important fact has been omitted throughout the paper, and that is that the level of geopolymerization that will take place largely depends on the amount of amorphous matter in the initial material(s).
- The text is poorly written in terms that the story flow was not properly followed, some key sentences are sometimes duplicated, some sentences do not show any clear fact, and are just theoretical claims, some of them are not placed in the right place in the text. It is very hard to read, and also very hard to explain all of the these-like issues. Therefore, please reread and rephrase the whole text with the highest concentration. Bear in mind not to list so many references for some claim, especially if that is already known and not some high new finding. Some of the examples are shown below, but there are many more. I suggest you to improve as much as you can, and then another round of revision would be required.
- If we are talking about materials or the effects that material gives to the soil, then give a concrete example of what has been used in previous research.
- Which clay minerals are swelling minerals?
- What did you mean by “low-grade materials” in these terms? (line 70)
- What did you mean by “clay tendencies”? (line 43)
- What does this sentence tell us? “The improvement towards the properties will lead to achieving desired engineering properties [6,8,12,16,17,23].” Is that significant, and does it need so many references cited?
- This claim was already written somewhere in the previous text, just with fewer details. I would suggest you to delete the first, less detailed one. “Additionally, soil stabilization can be described as a collective term for any physical, chemical, or combination of those approaches used to enhance certain features of natural soil in order for it to meet the engineering requirements [3,4, 9,10,11,15,21,24].”
- “Mechanical stabilization of a material is typically accomplished by adding another substance…” What is meant by “substance” here? Be more precise. Is it a chemical substance or a building material?
- This claim is too broad, be more precise: how is this obtained, which material can be used, in which quantity, how is that material made or set up in practice? (“The stiffness and strength of the material will typically be less than those obtained through chemical stabilization and will
frequently be unsuitable for heavy traffic pavements [33,34].”)
- This sentence does not seem to be at the right place in the text “Additionally, geopolymerization is the process of polymerizing inorganic natural materials to form geopolymers [43,56,57].”, and this word “additionally” should be deleted (lines 90-91).
- Rephrase, what is produced here? “Sources of alumina and silica act as precursors that are easily dissolved in alkaline solutions and produced by alkaline or silicate activation, making geopolymerization possible [47,49,57,58,60-63].” (lines 95-96)
- Precise which compressive strength you thought of here: “Calcium (CaO) content in GGBFS plays a role in the development of compressive strength [48,54,66,73,82,83,85,88].” (lines 251-252).
- Silica is there, it is not produced, but dissolves, reformulate “The reaction of sodium silicate (Na2SiO3) to process polymerization is crucial in producing more (Si), and the mixing ratio of sodium hydroxide (NaOH) concentration is also crucial in producing good strength of the product from geopolymer [48,51,75,82].”
-Change “has yielded high-strength results” to “promising results”.
- Change “tabulated in Table” to “presented in Table”.
- What is meant as impossible here? “As the solid to liquid ratio rose, the geopolymers sample became less homogeneous, which was impossible due to the limited amount of alkali activator [48,51,72,74,78,93,98].”
- Please:
- Change “fly ash based” in the whole text to “fly ash-based”
- Change “Clay soil” to “clayey soil”
- Change “is more significant than” to “higher than” (line 44)
- Change “stabilisation” to “stabilization” (and “stabilize”)
- Change “Stabilization of soils using chemical additions…” to “Stabilization of soils using chemical additives…” (line 80)
- “However, care must be exercised in selecting the proper compounds [25-27,39,40,45,47].” ??? (line 80)
- Change “minerals soil” to “minerals in soil” (lines 106-107)
- In lines 118-124, the first two sentences basically say the same thing, and the last sentence doesn't feel like belonging there.
- Change “rise” to “rising” (line 121)
- Delete the sentence (it does not belong there): “The unconfined compressive strength of the soil should be greater than 0.8 MPa. To increase compressive strength, adding other materials such as GGBFS may be necessary.“ in lines136-138.
- Change “Figure 1: Geopolymer Mechanism” to “Figure 1: Geopolymerization Mechanism”
- Abstract section should provide the information in which direction should the future research go.
- Some of the paragraphs contain plagiarism. Please, rewrite them.
- Lines 74-83
- Lines 238-252
- The whole text in 3.2. Alkali Activator.
- The title “Materials and methods” is wrong, because that still was the literature survey on the materials intended for soil stabilization.
6. Please, separate the use of FA and GGBFS in a traditional way, and by geopolymerization. Write the literature survey in that order, do not mix. First, give examples of pure FA with disadvantages and advantages, then examples of GGBFS (with disadvantages and advantages). Then sum up for the mixes.
Not every paragraph should explain the findings from every available paper, can you summarize, because that is the basic task of the review paper.
- There is no need to repeat the minimum limit for the compressive strength of 0.8 MPa.
I would suggest you make some figures comparing UCS and the content of FA and GGBFS, curing conditions, and activator concentration. You can then draw a line at 0.8 MPa, and the results would be clearly visible.
Also, I would suggest the diagrams concerning the other important parameters such as S/L.
You may think of how to organize these data to be visually comparable.
Author Response
Dear Reviewer,
Thanks for your comments and suggestions.
|
No. |
Comments |
Response |
|
1 |
English language and style (x) Extensive editing of English language and style required |
Has been sent for proof read again. |
|
2 |
Does the introduction provide sufficient background and include all relevant references? Comments: Must be improved |
Thanks for your comments. Please refer Introduction section, Lines 39-108. |
|
3 |
Is the research design appropriate? Comments: Must be improved |
Thanks for your comments. All changes are highlighted with blue colour fonts. |
|
4 |
Are the methods adequately described? Comments: Can be improved |
Thanks for your comments. All changes are highlighted with blue colour fonts. Please refer to the Sections 3 and 4, Lines 220-441. |
|
5 |
Are the results clearly presented? Comments: Must be improved |
Has been improved. Please refer to the Sections 3 and 4, Lines 220-441. All changes are highlighted with the blue colour fonts. |
|
6 |
Are the conclusions supported by the results? Comments: Must be improved |
Has been improved. Please refer to the Pages 14-15, Lines 443-488. |
|
7 |
If we are talking about materials or the effects that material gives to the soil, then give a concrete example of what has been used in previous research. |
Thanks for your comments. From the previous researches, the soil stabilization with fly ash and ground granulated blast furnace slag (GGBFS) based geopolymerization process could produce a better strength (> 0.8 Mpa and 80%) hence proven effective in increasing the strength of the soil as stated in ASTM D 4609 and Design Guideline for Alternative Pavement Structures (Low Volume Roads) of Malaysia Public Work Department (PWD). Please refer to the section 3.3, Page 12, Lines 340-354. |
|
8 |
Which clay minerals are potential swelling? |
Thanks for your comments. Kaolinite and muscovite are two clay minerals that are swelling minerals. Due to the fact that the primary clay minerals are kaolinite and muscovite, they have the potential to interact and absorb water. This results in a high liquid limit and plasticity index. Please refer to the section 1. Page 2, Lines 52-58. |
|
9 |
Line 70: What did you mean by “low-grade materials” in these terms?
|
Thanks for your comments. The term “low-grade materials” is soil with low strength, such as clay soil. In this new version, this term has been changed to problematic soil. Please refer to the section 1. Page 2, Lines 77-79. |
|
10 |
Line 43: What did you mean by “clay tendencies”? |
Thanks for your comments. The term “clay tendencies” is this soil the soil as to be classified as clay soil with high plasticity (ASTM D 2487). Please refer to the Page 2, Lines 47-48.
|
|
11 |
Line 56: What does this sentence tell us? “The improvement towards the properties will lead to achieving desired engineering properties [6,8,12,16,17,23].” Is that significant, and does it need so many references cited? |
Thanks for your comments. The sentence has been updated in this new version of manuscript. Please refer to the Page 2, Lines 64-65. |
|
12 |
Lines 57: This claim was already written somewhere in the previous text, just with fewer details. I would suggest you to delete the first, less detailed one. “Additionally, soil stabilization can be described as a collective term for any physical, chemical, or combination of those approaches used to enhance certain features of natural soil in order for it to meet the engineering requirements. |
Thanks for your comments and suggestions. The manuscript has been revised accordingly. The first similar statement has been deleted and the sentence at Page 2, Lines 66-68.
|
|
13 |
Line 63: “Mechanical stabilization of a material is typically accomplished by adding another substance…” What is meant by “substance” here? Be more precise. Is it a chemical substance or a building material? |
Thanks for your comments. The examples of the "substance" material have been added in the sentence. Please refer to the Page 2, Lines 71-73. |
|
14 |
Line 70-72: This claim is too broad, be more precise: how is this obtained, which material can be used, in which quantity, how is that material made or set up in practice? (“The stiffness and strength of the material will typically be less than those obtained through chemical stabilization and will frequently be unsuitable for heavy traffic pavements [33,34].”) |
Thanks for your comments. According to previous researches, the materials that can be utilized for soil stabilization include fly ash type C and GGBFS with a specified composition of 30% fly ash and 10%–30% GGBFS. The soil sample was prepared by mixing it with fly ash and GGBFS and curing it for 28 days. The unconfined compressive strength test (UCS) model is utilized as a practical indicator to determine the fly ash-GGBFS strength. Please refer to the section 3.3. Page 13, Lines 362-377. |
|
15 |
Lines 90-91: This sentence does not seem to be at the right place in the text “Additionally, geopolymerization is the process of polymerizing inorganic natural materials to form geopolymers [43,56,57].”, and this word “additionally” should be deleted. |
Thanks for your comments and suggestions. The manuscript has been revised. The sentence ‘’Geopolymerization is the process of polymerizing inorganic natural materials to form geopolymers [43,56,57]’’ has been reallocated and the word “additionally” has been deleted. Please refer to the Page 2, Lines 95-96. |
|
16 |
Lines 95-96: Rephrase, what is produced here? “Sources of alumina and silica act as precursors that are easily dissolved in alkaline solutions and produced by alkaline or silicate activation, making geopolymerization possible” |
Thanks for your comments and suggestions. The manuscript has been revised accordingly. Please refer to the Page 3, Lines 103-106. |
|
17 |
Lines 251-252: Precise which compressive strength you thought of here: “Calcium (CaO) content in GGBFS plays a role in the development of compressive strength [48,54,66,73,82,83,85,88].” |
Thanks for your comments The existence of calcium (CaO) content in GGBFS contributed to the shortened setting time and development of the compressive strength of the clay soil. The reaction between GGBFS and alkali activator solution forms a Calcium Aluminate Silicate Hydrate (C-A-S-H) gel-forming within the geopolymer matrix. These hydration products, along with aluminosilicate structure in the GGBFS samples, contributed to significantly gaining high strength. Please refer to the Page 9, Lines 268-273. |
|
18 |
Line 253: Silica is there, it is not produced, but dissolves, reformulate “The reaction of sodium silicate (Na2SiO3) to process polymerization is crucial in producing more (Si), and the mixing ratio of sodium hydroxide (NaOH) concentration is also crucial in producing good strength of the product from geopolymer [48,51,75,82]. |
Thanks for your comments and suggestions. We have revised and improved the manuscript accordingly in the new version. Please refer to the Page 9, Lines 283-286. |
|
19 |
Change “has yielded high-strength results” to “promising results”.
|
Thanks for your comments and suggestions. The sentence has been revised according to suggestion. Please refer to the Page 9, Line 300. |
|
20 |
Change “tabulated in Table” to “presented in Table”.
|
Thanks for your comments and suggestions. The manuscript has been revised as suggestion. Please refer to the section 2.3 Page 4, Lines 141-142. Please refer to the section 3.3 Page 9, Lines 301-302. |
|
21 |
Line 364: What is meant as impossible here? “As the solid to liquid ratio rose, the geopolymers sample became less homogeneous, which was impossible due to the limited amount of alkali activator [48,51,72,74,78,93,98].” |
Thanks for your comments The whole manuscript has been revised and improved accordingly. Please refer to the Page 13, Lines 398-399. |
|
22 |
Change “fly ash based” in the whole text to “fly ash-based” |
Thanks for your comments and suggestions. The whole text in the manuscript has been revised accordingly. |
|
23 |
Change “clay soil” to “clayey soil” |
Thanks for your comments and suggestions. The whole text in the manuscript has been revised accordingly. |
|
24 |
Line 44: Change “is more significant than” to “higher than”
|
Thanks for your comments. The statement of “is more significant than” has been revised accordingly as suggested. Please refer to the Page 2, Line 49. |
|
25 |
Change “stabilisation” to “stabilization” (and “stabilize”) |
Thanks for your comments. The words “stabilisation” and “stabilise” have been changed accordingly in the whole text in this new manuscript. |
|
26 |
Line 80: Change “Stabilization of soils using chemical additions…” to “Stabilization of soils using chemical additives…” (line 80)
|
Thanks for your comments. The sentence of ‘“Stabilization of soils using chemical additions…” has been revised as suggested. Please refer to the Page 2, Lines 89-91. |
|
27 |
Lines 106-107: Change “minerals soil” to “minerals in soil”
|
Thanks for your comments. The statement of ‘minerals soil’ has been revised. Please refer to the Page 3, Lines 113-114. |
|
28 |
Lines 118-124: the first two sentences basically say the same thing, and the last sentence doesn't feel like belonging there. |
Thanks for your comments. The sentence has been revised. Please refer to the Page 4, Lines 136-138. |
|
29 |
Line 137: Change “rise” to “rising”
|
Thanks for your comments. The statement of ‘rise’ has been revised as suggested. Please refer to the Page 4, Line 139. |
|
30 |
Lines 136-138: Delete the sentence (it does not belong there): “The unconfined compressive strength of the soil should be greater than 0.8 MPa. To increase compressive strength, adding other materials such as GGBFS may be necessary.“ |
Thanks for your comments. We have revised and improved the manuscript accordingly in the new version. Please refer to Page 4, Lines 164-169. |
|
31 |
Change “Figure 1: Geopolymer Mechanism” to “Figure 1: Geopolymerization Mechanism”
|
Thanks for your comments and suggestions. In new manuscript, Figure 1 become Figure 3 and the caption has been changed as suggested by reviewer. Please refer to the Page 8, Figure 3, Line 236. |
|
32 |
Abstract section should provide the information in which direction should the future research go.
|
Thanks for your comments and suggestions. The abstract has been revised and improved accordingly. Please refer to the Abstract section, Page 1, Lines 29-32 and Lines 35-36. |
|
33 |
Some of the paragraphs contain plagiarism. Please, rewrite them. - Lines 74-83 - Lines 238-252 - The whole text in 3.2. Alkali Activator. |
Thanks for your comments and suggestions. We have revised the manuscript accordingly in the new version. Please refer to the section 1, Page 2, Lines 77-91. Please refer to the section 3, Page 8, Lines 237-253. Please refer to the section 3.2, Page 9, Lines 275-294. |
|
34 |
The title “Materials and methods” is wrong, because that still was the literature survey on the materials intended for soil stabilization. |
Thanks for your comments. The title has been changed accordingly in the new version. Please refer to the section 2, Page 3, Line 109. |
|
35 |
Please, separate the use of FA and GGBFS in a traditional way, and by geopolymerization. |
Thanks for your comments and suggestions. The manuscript has been separated accordingly as suggested. Please refer to the Section 2, Pages 3-7, Lines 109-215. Please refer to the Section 3.3 Pages 9-13, Lines 295-381. |
Reviewer 3 Report
- Many references are cited in the article, and the classification is clear, which is worthy of recognition.
- There are many types of fly ash, and the article should clearly define the type of fly ash.
- Many research indicates that when Ground Granulated Blast Furnace Slag (GGBFS) mix with alkaline solution, geopolymer reactions and other reactions will exist simultaneously. It should be pointed out in the article.
- When using geopolymer technology in Soil Stabilization, there are generally have other reasons, ex: immobilizing heavy metals. It should be shown in the article.
- The performance of using geopolymer technology in Soil stabilization only discusses the compressive strength. The swell behavior, flexural strength, or abrasion resistance should be considered to discuss.
- Is it possible to evaluate and discuss the CO2 emission reduction ratio of Soil-based geopolymer?
Author Response
Dear Reviewer,
Thanks for your comments and suggestions.
|
No. |
Comments |
Response |
|
1 |
English language and style: (x) I don't feel qualified to judge about the English language and style |
Thanks for your comments. |
|
2 |
Does the introduction provide sufficient background and include all relevant references? Comment: Can be improved |
Thanks for your comments. Please refer Introduction section, Lines 39-108. |
|
3 |
Is the research design appropriate? Comment: Can be improved |
Thanks for your comments. All changes are highlighted with blue colour fonts. |
|
4 |
Are the methods adequately described? Comment: Yes |
Thanks for your comments. |
|
5 |
Are the results clearly presented? Comment: Can be improved |
Has been improved. Please refer to the Sections 3 and 4, Lines 220-441. All changes are highlighted with the blue colour fonts. |
|
6 |
Are the conclusions supported by the results? Comment: Can be improved |
Has been improved. Please refer to the Pages 14-15, Lines 443-488. |
|
7 |
There are many types of fly ash, and the article should clearly define the type of fly ash.
|
Thanks for your comments. In this paper, the soil stabilization based geopolymer using fly ash Class C. Please refer to the Introduction section, page 3, lines 106-108. Please refer to the Summary and future works section, pages 14-15, lines 443-475. |
|
8 |
Many research indicates that when Ground Granulated Blast Furnace Slag (GGBFS) mix with alkaline solution, geopolymer reactions and other reactions will exist simultaneously. It should be pointed out in the article.
|
Thanks for your comments. Previous researches indicates that when Ground Granulated Blast Furnace Slag (GGBFS) mix with alkaline solution, geopolymer reactions will exist simultaneously have been added in this new manuscript. Please refer to the Section 3.3, Page 13, Lines 371-377. |
|
9 |
When using geopolymer technology in Soil Stabilization, there are generally have other reasons?
|
Thanks for your comments. Geopolymers have outstanding engineering properties, such as increased strength and improved soil adhesion. Please refer to the Section 1, Page 2, Lines 98-99.
|
|
10 |
The performance of using geopolymer technology in Soil stabilization only discusses the compressive strength. The swell behavior, flexural strength, or abrasion resistance should be considered to discuss. |
Thanks for your comments and suggestions. The swell behaviour, flexural strength, or abrasion resistance of soil stabilization based geopolymer using GGBFS, and fly ash Class C in road construction, particularly in the subgrade layer of the soil has not been investigated. Thus, this article only discusses the performance of using geopolymer technology in soil stabilization based on compressive strength in road construction applications, especially in subgrade layers. Please refer to the Summary and future works section, Page 15, Lines 468-473. |
|
11 |
Is it possible to evaluate and discuss the CO2 emission reduction ratio of Soil-based geopolymer?
|
Thanks for your comments. If the soil stabilization only uses geopolymers, it does not produce CO2. hence, it is impossible to evaluate and discuss the CO2 emission reduction ratio of Soil-based geopolymers. However, if soil stabilization involves the use of materials such as cement, it is possible to evaluate and discuss the CO2 emission. Because one ton of cement is manufactured, accounting for approximately one ton of greenhouse gas CO2 released into the atmosphere as a result of lime decarbonisation in the kiln during cement manufacture. Please refer to the Summary and future works section, Page 15, Lines 461-468. |
Round 2
Reviewer 2 Report
The Manuscript has been much improved now.
There are still a few remarks.
- Kaolinite is not a swelling clay.
- Fig. 3 is not clear, the resolution is low.
Author Response
Dear Reviewer,
|
Thanks for your comments. We have really appreciated it. |
|
No. |
Comments |
Response |
|
1 |
Kaolinite is not a swelling clay mineral. |
Thanks for your comments. We are really appreciated it. The sentences have been revised accordingly. Please refer to the Page 2, Lines 52-54. |
|
2 |
Fig. 3 is not clear; the resolution is low.
|
Thanks for your comments. The quality of Figure 3 has been upgraded accordingly. Please refer to Figure 3, Page 8. |